# Batch Multi-Fidelity Active Learning with Budget Constraints

**Shibo Li**\*, **Jeff M. Phillips**\*, **Xin Yu, Robert M. Kirby, and Shandian Zhe**
School of Computing, University of Utah
Salt Lake City, UT 84112
`{shibo, jeffp, xiny, kirby, zhe}@cs.utah.edu`

## Abstract

Learning functions with high-dimensional outputs is critical in many applications, such as physical simulation and engineering design. However, collecting training examples for these applications is often costly, *e.g.*, by running numerical solvers. The recent work (Li et al., 2022) proposes the first multi-fidelity active learning approach for high-dimensional outputs, which can acquire examples at different fidelities to reduce the cost while improving the learning performance. However, this method only queries at one pair of fidelity and input at a time, and hence has a risk to bring in strongly correlated examples to reduce the learning efficiency. In this paper, we propose Batch Multi-Fidelity Active Learning with Budget Constraints (BMFAL-BC), which can promote the diversity of training examples to improve the benefit-cost ratio, while respecting a given budget constraint for batch queries. Hence, our method can be more practically useful. Specifically, we propose a novel batch acquisition function that measures the mutual information between a batch of multi-fidelity queries and the target function, so as to penalize highly correlated queries and encourages diversity. The optimization of the batch acquisition function is challenging in that it involves a combinatorial search over many fidelities while subject to the budget constraint. To address this challenge, we develop a weighted greedy algorithm that can sequentially identify each (fidelity, input) pair, while achieving a near $(1 - 1/e)$-approximation of the optimum. We show the advantage of our method in several computational physics and engineering applications.

## 1 Introduction

Applications, such as in computational physics and engineering design, often demand we calculate a complex mapping from low-dimensional inputs to high-dimensional outputs, such as finding the optimal material layout (output) given the design parameters (input), and solving the solution field on a mesh (output) given the PDE parameters (input). Computing these mappings is often very expensive, *e.g.*, iteratively running numerical solvers. Hence, learning a surrogate model to outright predict the mapping, which is much faster and cheaper, is of great practical interest and importance (Kennedy and O'Hagan, 2000; Conti and O'Hagan, 2010)

However, collecting training examples for the surrogate model becomes another bottleneck, since each example still requires a costly computation. To alleviate this issue, Li et al. (2022) developed DMFAL, a first deep multi-fidelity active learning algorithm, which can acquire examples at different fidelities to reduce the cost of data collection. Low-fidelity examples are cheap to compute (*e.g.*, with coarse meshes) yet inaccurate; high-fidelity examples are accurate but much more expensive (*e.g.*, calculated with dense grids). See Fig. 1 for an illustration. DMFAL uses an optimization-based acquisition method to dynamically identify the input and fidelity at which to query new examples, so as to improve the learning performance, lower the sample complexity, and reduce the cost.

---

\*Equal contribution    Correspondence to: Jeff M. Phillips, Shandian Zhe.

36th Conference on Neural Information Processing Systems (NeurIPS 2022).

Despite its effectiveness, DMFAL can only optimize and query at one pair of input and fidelity each time and hence ignores the correlation between consecutive queries. As a result, it has a risk of bringing in strongly correlated examples, which can restrict the learning efficiency and lead to a suboptimal benefit-cost ratio. In addition, the sequential querying and training strategy is difficult to utilize parallel computing resources that are common nowadays (*e.g.*, multi-core CPUs/GPUs and computer clusters) to query concurrently and to further speed up.

In this paper, we propose BMFAL-BC, a batch multi-fidelity active learning method with budget constraints. Our method can acquire a batch of multi-fidelity examples at a time to inhibit the example correlations, promote diversity so as to improve the learning efficiency and benefit-cost ratio. Our method can respect a given budget in issuing batch queries, hence are more widely applicable and practically useful. Specifically, we first propose a novel acquisition function, which measures the mutual information between a batch of multi-fidelity queries and the target function. The acquisition function not only can penalize highly correlated queries to encourage diversity, but also can be efficiently computed by an Monte-Carlo approximation. However, optimizing the acquisition function is challenging because it incurs a combinatorial search over fidelities and meanwhile needs to obey the constraint. To address this challenge, we develop a weighted greedy algorithm. We sequentially find one pair of fidelity and input each step, by maximizing the increment of the mutual information weighted by the cost. In this way, we avoid enumerating the fidelity combinations and greatly improve the efficiency. We prove that our greedy algorithm nearly achieves a $(1 - \frac{1}{e})$-approximation of the optimum, with a few minor caveats.

For evaluation, we examined BMFAL-BC in five real-world applications, including three benchmark tasks in physical simulation (solving Poisson's, Heat and viscous Burger's equations), a topology structure design problem, and a computational fluid dynamics (CFD) task to predict the velocity field of boundary-driven flows. We compared with the budget-aware version of DMFAL, single multi-fideity querying with our acquisition function, and several random querying strategies. Under the same budget constraint, our method consistently outperforms the competing methods throughout the learning process, often by a large margin.

## 2 Background

### 2.1 Problem Setting

Suppose we aim to learn a mapping $f : \Omega \subseteq \mathbb{R}^r \to \mathbb{R}^d$ where $r$ is small but $d$ is large, *e.g.*, hundreds of thousands. To economically learn this mapping, we collect training examples at $M$ fidelities. Each fidelity $m$ corresponds to mapping $f_m : \Omega \to \mathbb{R}^{d_m}$. The target mapping is computed at the highest fidelity, *i.e.*, $f(\mathbf{x}) = f_M(\mathbf{x})$. The other $f_m$ can be viewed as a (rough) approximation of $f$. Note that $d_m$ is unnecessarily the same as $d$ for $m < M$. For example, solving PDEs on a coarse mesh will give a lower-dimensional output (on the mesh points). However, we can interpolate it to the $d$-dimensional space to match $f(\cdot)$ (this is standard in physical simulation (Zienkiewicz et al., 1977)). Denote by $\lambda_m$ the cost of computing $f_m(\cdot)$ at fidelity $m$. We have $\lambda_1 \leq \ldots \leq \lambda_M$.

### 2.2 Deep Multi-Fidelity Active Learning (DMFAL)

To effectively estimate $f$ while reducing the cost, Li et al. (2022) proposed DMFAL, a multi-fidelity deep active learning approach. Specifically, a neural network (NN) is introduced for each fidelity $m$, where a low-dimensional hidden output $\mathbf{h}_m(\mathbf{x})$ is first generated, and then projected to the high-dimensional observation space. Each NN is parameterized by $(\mathbf{A}_m, \mathbf{W}_m, \boldsymbol{\theta}_m)$, where $\mathbf{A}_m$ is the projection matrix, $\mathbf{W}_m$ is the weight matrix of the last layer, and $\boldsymbol{\theta}_m$ consists of the remaining NN parameters. The model is defined as follows,

$$\mathbf{x}_m = [\mathbf{x}; \mathbf{h}_{m-1}(\mathbf{x})], \quad \mathbf{h}_m(\mathbf{x}) = \mathbf{W}_m \boldsymbol{\phi}_{\boldsymbol{\theta}_m}(\mathbf{x}_m), \quad \mathbf{y}_m(\mathbf{x}) = \mathbf{A}_m \mathbf{h}_m(\mathbf{x}) + \boldsymbol{\xi}_m, \tag{1}$$

where $\mathbf{x}_m$ is the input to the NN at fidelity $m$, $\mathbf{y}_m(\mathbf{x})$ is the observed $d_m$ dimensional output, $\boldsymbol{\xi}_m \sim \mathcal{N}(\cdot | \mathbf{0}, \tau_m \mathbf{I})$ is a random noise, $\boldsymbol{\phi}_{\boldsymbol{\theta}_m}(\mathbf{x}_m)$ is the output of the second last layer and can be viewed as a nonlinear feature transformation of $\mathbf{x}_m$. Since $\mathbf{x}_m$ includes not only the original input $\mathbf{x}$, but also the hidden output from the previous fidelity, *i.e.*, $\mathbf{h}_{m-1}(\mathbf{x})$, the model can propagate information throughout fidelities and capture the complex relationships (*e.g.*, nonlinear and nonstationary) between different fidelities. The whole model is visualized in Fig. 5 of Appendix. To estimate the posterior of the model, DMFAL uses a structural variational inference algorithm. A multi-variate Gaussian

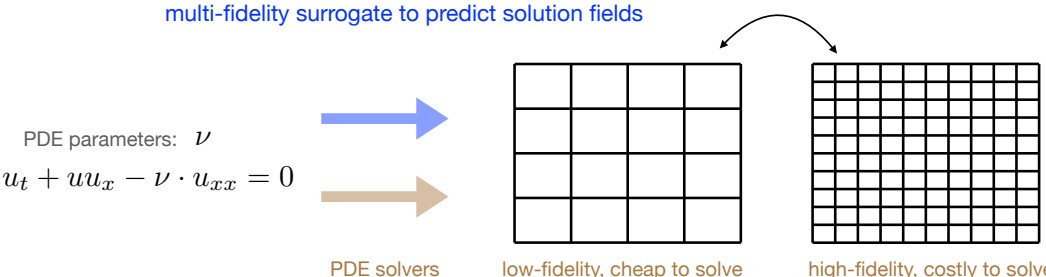

multi-fidelity surrogate to predict solution fields

PDE parameters: $\nu$

$$u_t + uu_x - \nu \cdot u_{xx} = 0$$

PDE solvers    low-fidelity, cheap to solve    high-fidelity, costly to solve

Figure 1: Illustration of the motivation and goal with physical simulation as an example. It is costly to solve every PDE from scratch. We aim to train a multi-fidelity surrogate model to directly predict high-fidelity solution fields given the PDE parameters. To further reduce the cost of collecting the training data with numerical solvers, we seek to develop multi-fidelity active learning algorithms.

posterior is introduced for each weight matrix, $q(\mathbf{W}_m) = \mathcal{N}\big(\text{vec}(\mathbf{W}_m)|\boldsymbol{\mu}_m, \boldsymbol{\Sigma}_m\big)$. A variational evidence lower bound (ELBO) is maximized via stochastic optimization and the reparameterization trick (Kingma and Welling, 2013).

To conduct active learning, DMFAL views the most valuable example at each fidelity $m$ as the one that can best help its prediction at the highest fidelity $M$ (*i.e.*, the target function). Accordingly, the acquisition function is defined as

$$a(m, \mathbf{x}) = \frac{1}{\lambda_m}\mathbb{I}\big(\mathbf{y}_m(\mathbf{x}), \mathbf{y}_M(\mathbf{x})|\mathcal{D}\big) = \frac{1}{\lambda_m}\left(\mathbb{H}(\mathbf{y}_m|\mathcal{D}) + \mathbb{H}(\mathbf{y}_M|\mathcal{D}) - \mathbb{H}(\mathbf{y}_m, \mathbf{y}_M|\mathcal{D})\right), \quad (2)$$

where $\mathbb{I}(\cdot, \cdot)$ is the mutual information, $\mathbb{H}(\cdot)$ is the entropy, and $\mathcal{D}$ is the current training dataset. The computation of the acquisition function is quite challenging because $\mathbf{y}_m$ and $\mathbf{y}_M$ are both high dimensional. To address this issue, DMFAL takes advantage of the fact that each low-dimensional output $\mathbf{h}_m(\mathbf{x})$ is a nonlinear function of the random weight matrices $\{\mathbf{W}_1, \ldots, \mathbf{W}_m\}$. Based on the variational posterior $\{q(\mathbf{W}_j)\}$, DMFAL uses the multi-variate delta method (Oehlert, 1992; Bickel and Doksum, 2015) to estimate the mean and covariance of $\widehat{\mathbf{h}} = [\mathbf{h}_m; \mathbf{h}_M]$, with which to construct a joint Gaussian posterior approximation for $\widehat{\mathbf{h}}$ by moment matching. Then, from the projection in (1), we can derive a Gaussian posterior for $[\mathbf{y}_m; \mathbf{y}_M]$. By further applying Weinstein-Aronszajn identify (Kato, 2013), we can compute the entropy terms in (2) conveniently and efficiently.

Each time, DMFAL maximizes the acquisition function (2) to identify a pair of input and fidelity at which to query. The new example is added into $\mathcal{D}$, and the deep multi-fidelity model is retrained and updated. The process repeats until the maximum number of training examples have been queried or other stopping criteria are met.

## 3 Batch Multi-Fidelity Active Learning with Budget Constraints

While effective, DMFAL can only identify and query at one input and fidelity each time, hence ignoring the correlations between the successive queries. As a result, highly correlated examples are easier to be acquired and incorporated into the training set. Consider we have found $(\mathbf{x}^*, m^*)$ that maximizes the acquisition function (2). It is often the case that a single example will not cause a significant change of the model posterior $\{q(\mathbf{W}_m)\}$ (especially when the current dataset $\mathcal{D}$ is not very small). When we optimize the acquisition function again, we are likely to obtain a new pair $(\hat{\mathbf{x}}^*, \hat{m}^*)$ that is close to $(\mathbf{x}^*, m^*)$ (*e.g.*, $\hat{m}^* = m^*$ and $\hat{\mathbf{x}}^*$ close to $\mathbf{x}^*$). Accordingly, the queried example will be highly correlated to the previous one. The redundant information within these examples can restrict the learning efficiency, and demand for more queries (and cost) to achieve the satisfactory performance. Note that the similar issue have been raised in other single-fidelity, pool-based active learning works, *e.g.*, (Geifman and El-Yaniv, 2017; Kirsch et al., 2019); see Sec 4.

To overcome this problem, we propose BMFAL-BC, a batch multi-fidelity active learning approach to reduce the correlations and to promote the diversity of training examples, so that we can improve the learning performance, lower the sample complexity, and better save the cost. In addition, we take into account the budget constraint in querying the batch examples, which is common in practice (like cloud or high-performance computing) (Mendes et al., 2020). Under the given budget, we aim to find a batch of multi-fidelity queries that improves the benefit-cost ratio as much as possible.

## 3.1 Batch Acquisition Function

Specifically, we first consider an acquisition function that allows us to jointly optimize a set of inputs and fidelities. While it seems natural to consider how to extend (2) to the batch case, the acquisition function in (2) is about the mutual information between $\mathbf{y}_m(\mathbf{x})$ and $\mathbf{y}_M(\mathbf{x})$. That means, it only measures the utility of the query $(m, \mathbf{x})$ in improving the estimate of the target function at $\mathbf{x}$ itself (*i.e.*, $\mathbf{y}_M(\mathbf{x})$), rather than at any other input. To take into account the utility in improving the function estimation at all the inputs, we therefore propose a new single acquisition function,

$$a_s(m, \mathbf{x}) = \mathbb{E}_{p(\mathbf{x}')}\left[\mathbb{I}\left(\mathbf{y}_m(\mathbf{x}), \mathbf{y}_M(\mathbf{x}')|\mathcal{D}\right)\right] \tag{3}$$

where $\mathbf{x}' \in \Omega$ and $p(\mathbf{x}')$ is the distribution of the input in $\Omega$. We can see that, by varying the input $\mathbf{x}'$ in the second argument of the mutual information, we are able to evaluate the utility of the query in improving the estimation of the entire body of the target function. Hence, it is more reasonable and comprehensive. Now, consider querying a batch of examples under the budget $B$, we extend (3) to

$$a_{\text{batch}}(\mathcal{M}, \mathcal{X}) = \mathbb{E}_{p(\mathbf{x}')}\left[\mathbb{I}\left(\{\mathbf{y}_{m_j}(\mathbf{x}_j)\}_{j=1}^n, \mathbf{y}_M(\mathbf{x}')|\mathcal{D}\right)\right], \quad \text{s.t.} \ \sum_{j=1}^n \lambda_{m_j} \leq B, \tag{4}$$

where $\mathcal{M} = \{m_1, \ldots, m_n\}$, $\mathcal{X} = \{\mathbf{x}_1, \ldots, \mathbf{x}_n\}$, and $\mathcal{D}$ is the current training dataset. We can see that the more correlated $\{\mathbf{y}_{m_j}(\mathbf{x}_j)\}_{j=1}^n$, the smaller the mutual information, and hence the smaller the expectation in (4). Therefore, the batch acquisition function implicitly penalizes strongly correlated queries and favors diversity.

The expected mutual information in (4) is usually analytically intractable. However, we can efficiently compute it with an Monte-Carlo approximation. That is, we draw $A$ independent samples from the input distribution, $\mathbf{x}'_1, \ldots, \mathbf{x}'_A \sim p(\mathbf{x}')$, and compute

$$\widehat{a}_{\text{batch}}(\mathcal{M}, \mathcal{X}) = \frac{1}{A} \sum_{l=1}^A \mathbb{I}\left(\{\mathbf{y}_{m_j}(\mathbf{x}_j)\}_{j=1}^n, \mathbf{y}_M(\mathbf{x}'_l)|\mathcal{D}\right). \tag{5}$$

It is straightforward to extend the multi-variate delta method used in DMFAL to calculate the mutual information in (5). We can then maximize (5) subject to the budget constraint, $\sum_{j=1}^B \lambda_{m_j} \leq B$, to acquire more diverse and informative training examples. In addition, when parallel computing resources (*e.g.*, multi-core CPUs/GPUs and computer clusters) are available, we can acquire these queries in parallel to further speed up the active learning.

However, directly maximizing (5) will incur a combinatorial search over multiple fidelities in $\mathcal{M}$, and hence is very expensive. Note that the number of examples $n$ is not fixed, and can vary as long as the cost does not exceed the budget $B$, which makes the optimization even more challenging. To address these issues, we propose a weighted greedy algorithm to sequentially determine each $(m_j, \mathbf{x}_j)$ pair.

## 3.2 Weighted Greedy Optimization

Specifically, at each step, we maximize the mutual information increment on one pair of input and fidelity, weighted by the corresponding cost. Suppose at step $k$, we have identified a set of $k$ inputs and fidelities at which to query, $\mathcal{Q}_k = \{(\mathbf{x}_j, m_j)\}_{j=1}^k$. To identify a new pair of input and fidelity at step $k + 1$, we maximize a weighted incremental version of (5),

$$\hat{a}_{k+1}(\mathbf{x}, m) = \frac{1}{A} \sum_{l=1}^A \frac{\mathbb{I}\left(\mathcal{Y}_k \cup \{\mathbf{y}_m(\mathbf{x})\}, \mathbf{y}_M(\mathbf{x}'_l)|\mathcal{D}\right) - \mathbb{I}\left(\mathcal{Y}_k, \mathbf{y}_M(\mathbf{x}'_l)|\mathcal{D}\right)}{\lambda_m}$$

$$\text{s.t.} \ \lambda_m + \sum_{\widehat{m} \in \mathcal{M}_k} \lambda_{\widehat{m}} \leq B, \tag{6}$$

where $\mathcal{Y}_k = \{\mathbf{y}_{m_j}(\mathbf{x}_j)|(\mathbf{x}_j, m_j) \in \mathcal{Q}_k\}$, and $\mathcal{M}_k$ are all the fidelities in $\mathcal{Q}_k$. At the beginning ($k = 0$), we have $\mathcal{Q}_k = \emptyset$, $\mathcal{Y}_k = \emptyset$ and $\mathcal{M}_k = \emptyset$. Each step, we look at each valid fidelity, *i.e.*, $1 \leq \lambda_m \leq B - \sum_{\widehat{m} \in \mathcal{M}_k} \lambda_{\widehat{m}}$, and find the optimal input. We then add the pair $(m, \mathbf{x})$ that gives the largest $\hat{a}_{k+1}$ into $\mathcal{Q}_k$, and proceed to the next step. Our greedy approach is summarized in Algorithm 1.

---

**Algorithm 1** Weighted-Greedy( $\{\lambda_m\}$, budget $B$)

---

1: $k \leftarrow 0, \mathcal{Q}_k \leftarrow \emptyset, C_k \leftarrow 0$
2: **while** $C_k \leq B$ **do**
3:     Optimize the weighted incremental acquisition function in (6):

$$(\mathbf{x}^*, m^*) = \operatorname*{argmax}_{\mathbf{x} \in \Omega, 1 \leq m \leq B - C_k} \hat{a}_{k+1}(\mathbf{x}, m)$$

4:     **if** Infeasible **then**
5:        break
6:     **end if**
7:     $k \leftarrow k + 1$
8:     $\mathcal{Q}_k \leftarrow \mathcal{Q}_{k-1} \cup \{(\mathbf{x}^*, m^*)\}$
9:     $C_k \leftarrow C_{k-1} + \lambda_{m^*}$
10: **end while**
11: Return $\mathcal{Q}_k$

---

The intuition of our approach is as follows. Mutual information is a classic submodular function (Krause and Guestrin, 2005), and hence if there were no budget constraints or weights, greedily choosing the input which increases the mutual information most achieves a solution within $(1 - 1/e)$ of the optimal (Krause and Golovin, 2014). However, Leskovec et al. (2007) observed that when items have a weight (corresponding to the cost for different fidelities in our case) and there is a budget constraint, then the approximation factor can be arbitrarily bad. We observe, *and prove*, that this only occurs as the budget is about to be filled, and up until that point, the weighted greedy optimization achieves the best possible $(1 - 1/e)$-approximation of the optimal. We can formalize this *near* $(1 - 1/e)$-*approximation* in two ways, proven in the Appendix. Let OPT($B$) be the maximum amount of mutual information that can be achieved with a budget $B$.

**Theorem 3.1.** *At any step of Weighted-Greedy (Algorithm 1) before any choice of fidelity would exceed the budget, and the total budget used to that point is $B' < B$, then the mutual information of the current solution is within $(1 - 1/e)$ of OPT($B'$).*

**Corollary 3.1.** *If Weighted-Greedy (Algorithm 1) is run until input-fidelity pair $(\mathbf{x}, m)$ that corresponds with the maximal acquisition function $\hat{a}_{k+1}(\mathbf{x}, m)$ would exceed the budget, it selects that input-fidelity pair anyways (the solution exceeds the budget $B$) and then terminates, the solution obtained is within $(1 - 1/e)$ of OPT($B$).*

We next sketch the main idea of how to prove the main result. Adding a new fidelity-input pair $(m, \mathbf{x})$ gives an increment of learning benefit $\Delta_j = \mathbb{I}(\mathcal{Y}_k \cup \{\mathbf{y}_m(\mathbf{x})\}, \mathbf{y}_M(\mathbf{x}'_l)|\mathcal{D}) - \mathbb{I}(\mathcal{Y}_k, \mathbf{y}_M(\mathbf{x}'_l)|\mathcal{D})$. Since we need to trade off to the cost $\lambda_m$, we can view there are $\lambda_m$ independent copies of $\mathbf{x}$ (for the particular fidelity $m$), and adding each copy gives $\frac{\Delta_j}{\lambda_m}$ benefit. We choose the optimal $\mathbf{x}$ and $m$ that maximizes the benefit $\frac{\Delta_j}{\lambda_m}$. Since all the $\lambda_m$ copies of $\mathbf{x}$ have the equal, biggest benefit (among all possible choices of inputs in $\Omega$ and fidelities in $\mathcal{M}$), we choose to add them first (greedy strategy), which in total gives $\Delta_j$ benefit – their benefit does not diminish as each copy is added. Via dividing by $\lambda_m$ and considering the copies, which each have unit weight, we can apply the standard submodular optimization analysis obtaining $(1 - 1/e)$OPT, at least until we encounter the budget constraint.

### 3.3 Algorithm Complexity

The time complexity of our weighted greedy optimization is $\mathcal{O}(\frac{B}{\lambda_1} MG)$ where $\lambda_1$ is the cost for the lowest fidelity, $G$ is the cost of the underlying numerical optimization (*e.g.*, L-BFGS) and acquisition function computation (detailed in (Li et al., 2022)). The space complexity is $\mathcal{O}(\frac{B}{\lambda_1}(r + d + 1))$, which is to store at most $B/\lambda_1$ pairs of input locations and fidelities, and their corresponding outputs (*i.e.*, the querying results). Therefore, both the time and space complexities are linear to the budget $B$.

## 4 Related Work

As an important branch of machine learning, active learning has been studied for a long time (Balcan et al., 2007; Settles, 2009; Balcan et al., 2009; Dasgupta, 2011; Hanneke et al., 2014). While many prior works develop active learning algorithms for kernel based models, *e.g.*, (Schohn and Cohn, 2000;

Tong and Koller, 2001; Joshi et al., 2009; Krause et al., 2008; Li and Guo, 2013; Huang et al., 2010), recent research focus has transited to deep neural networks. Gal et al. (2017) used Monte-Carlo (MC) Dropout (Gal and Ghahramani, 2016) to generate posterior samples of the neural network output, based on which a variety of acquisition functions, such as predictive entropy and Bayesian Active Learning by Disagreement (BALD) (Houlsby et al., 2011), can be calculated and optimized to query new examples in active learning. This approach has been proven successful in image classification. Kirsch et al. (2019) developed BatchBALD, a greedy approach that incrementally selects a set of unlabeled images under the BALD principle and issues batch queries to improve the active learning efficiency. They show that the batch acquisition function based on BALD is submodular, and hence the greedy approach achieves a $1 - 1/e$ approximation. Other works along this line includes (Geifman and El-Yaniv, 2017; Sener and Savarese, 2018) based on core-set search, (Gissin and Shalev-Shwartz, 2019; Ducoffe and Precioso, 2018) based on adversarial networks or samples, (Ash et al., 2019) based on the uncertainty evaluated in the gradient magnitude, *etc*.

Different from most methods, DMFAL (Li et al., 2022) conducts optimization-based, rather than pool-based active learning. That is, they optimize the acquisition function in the entire domain rather than rank a set of pre-collected examples to label. It is also related to Bayesian experimental design (Kleinegesse and Gutmann, 2020). In addition, DMFAL for the first time automatically queries examples at different fidelities, and integrates these examples in a deep multi-fidelity model to improve the active learning efficiency while reducing the cost — which is critical in physical simulation and engineering design. The pioneer work of Settles et al. (2008) empirically studies how the human labeling cost can vary in practical active learning, but does not provide a scheme to identify multi-fidelity queries. Our work is an extension of (Li et al., 2022) to generate a batch of multi-fidelity queries so as to reduce the query correlations, improve the diversity and quality of the training examples, while respecting a given budget constraint. A counterpart in the Bayesian optimization domain is the recent work BMBO-DARN (Li et al., 2021), which considers batch multi-fidelity queries for optimizing a black-box function. From the high-level view, BMBO-DARN and our work employ a similar interleaving procedure, *i.e.*, determining a new batch of queries by optimizing an acquisition function, issuing the queries to collect new examples, and updating the surrogate model. However, both the learning setting and acquisition function are different. More important, we consider the budget constraint while BMBO-DARN does not. Thereby, the computation and optimization techniques of the two works are totally different. The BMBO-DARN uses Hamiltonian Monte-Carlo samples of the single function output prediction and constructs empirical covariance matrices to compute the acquisition function, while our method and (Li et al., 2022) use the multi-variate delta method and matrix identities to overcome the challenge of the high-dimensional outputs. BMBO-DARN uses alternating optimization to search over multiple fidelities, while our work develops a weighted greedy algorithm with additional theoretical guarantees to respect the budget constraint.

## 5 Experiment

### 5.1 Solving Partial Differential Equations

We first evaluated BMFAL-BC in several benchmark tasks of computational physics, *i.e.*, predicting the solution fields of partial differential equations (PDEs), including *Heat*, *Poisson*'s, and *Burgers*' equations (Olsen-Kettle, 2011). A numerical solver was used to collect the training examples. High-fidelity examples were generated by running the solver with dense meshes, while low-fidelity examples by coarse meshes. The dimension of the output is the number of the grid points. For example, a $50 \times 50$ mesh means the output dimension is $2,500$. We provided two-fidelity queries for each PDE, corresponding to $16 \times 16$ and $32 \times 32$ meshes. We also tested three-fidelity queries for Poisson's equation, with a $64 \times 64$ mesh to generate examples at the third fidelity. We denote the three-fidelity setting by Poisson-3. The input comprises of the PDE parameters and/or boundary/initial condition parameters. The details are provided in (Li et al., 2022). We ran the solver at each fidelity for many times, and computed the average running time. We normalized the average running time to obtain the querying cost at each fidelity, $\lambda_1 = 1$, $\lambda_2 = 3$ and $\lambda_3 = 10$. To collect the initial training dataset for active learning, we generated 10 fidelity-1 examples and 2 fidelity-2 examples for in the two-fidelity setting, and 10, 5, and 2 examples of fidelity-1, 2, 3, respectively, for the three-fidelity setting. The training inputs of the initial dataset were uniformly sampled from the domain. To evaluate the prediction accuracy, for each PDE, we randomly sampled 500 inputs, calculated the ground-truth outputs from a much denser mesh: $128 \times 128$ for Burger's and Poisson's and $100 \times 100$

for Heat equation. The predictions at the highest fidelity were then interpolated to these denser meshes (Zienkiewicz et al., 1977) to evaluate the error.

**Competing Methods.** Since currently there is not any budget-aware, batch multi-fidelity active learning approach (to our knowledge), for comparison, we first made a simple extension of the state-of-the-art multi-fidelity active learning method, DMFAL (Li et al., 2022). Specifically, to obtain a batch of queries, we ran DMFAL as it is, namely, each step acquiring one example by maximizing (2) and then retraining the model, until the budget is exhausted or no new queries can be issued (otherwise the budget will be exceeded). We denote this method by (i) `DMFAL-BC`. Note that it is still sequentially querying and training inside each batch, but respects the budget. Next, to confirm the effectiveness of the proposed new acquisition function (3) (based on which, we propose our batch acquisition function (4)), we ran active learning in the same away as `DMFAL-BC`, except the acquisition function is replaced by $\frac{a_s(m,\mathbf{x})}{\lambda_m}$ where $a_s$ is defined by (3). To compute $a_s$, we used an Monte-Carlo approximation similar to (5), where the number of samples $A$ is 20. We denote this method by (ii) `MFAL-BC`. In addition, we compared with (iii) `DMFAL-BC-RF`, which follows the running of `DMFAL-BC`, but each time, it randomly selects a fidelity $m$, then maximize the mutual information $\mathbb{I}(\mathbf{y}_m(\mathbf{x}), \mathbf{y}_M(\mathbf{x})|\mathcal{D})$ — the numerator of (2) — to identify the corresponding input. Similarly, we tested (iv) `MFAL-BC-RF`, which follows the execution of `MFAL-BC`, but each time, it randomly selects a fidelity $m$ and maximizes $a_s(m, \mathbf{x})$. Again $a_s$ is computed by the Monte-Carlo approximation. For all these methods, we used L-BFGS for the input optimization. Finally, we tested (v) `Batch-FR-BC`, which randomly samples both the fidelity and input at each step (*i.e.*, fully random), until the budget is used up or no more fidelities are available. Then the batch of the acquired examples were added into the dataset altogether to retrain the model. Note that there are other possible acquisition functions, *e.g.*, the popular predictive variances and BALD (Houlsby et al., 2011). Their multi-fidelity versions have been tested and compared against DMFAL in (Li et al., 2022), and turns out to be inferior. Hence, we did not test their possible variants/extensions in our paper.

**Settings and Results.** All the methods were implemented by Pytorch (Paszke et al., 2019). We followed the same setting as in Li et al. (2022) to train the deep multi-fidelity model (see Sec. 2.2), which employed a two-layer NN at each fidelity, `tanh` activation, and the layer width was selected from $\{20, 40, 60, 80, 100\}$ from the initial training data. The dimension of the latent output was 20. The learning rate was tuned from $\{10^{-4}, 5 \times 10^{-4}, 10^{-3}, 5 \times 10^{-3}, 10^{-2}\}$. We set the budget for acquiring each batch to 20 (normalized seconds), and ran each method to acquire 25 batches of training examples. We tracked the running performance of each method in terms of the normalized root-mean-square-error (nRMSE). We repeated the experiment for five times, and report the average nRMSE *vs.* the accumulated cost (*i.e.*, the summation of the corresponding $\lambda$'s in each acquired example) in Fig. 2. The shaded region exhibits the standard deviation. As we can see, BMFAL-BC consistently outperforms all the competing methods by a large margin. At very early stages, all the methods exhibit similar prediction accuracy. This is reasonable, because they started with the same training set. However, along with more batches of queries, BMFAL-BC shows better performance, and its discrepancy with the baselines is in general growing. In addition, MFAL-BC constantly outperforms DMFAL-BC. Since they conduct the same one-by-one querying and training strategy, the improvement MFAL-BC reflects the advantage of our new single acquisition function (3) over the one used in DMFAL. Together these results have demonstrated the advantage of our method.

## 5.2   Topology Structure Optimization

Next, we applied our approach in topology structure optimization, which is critical in many manufacturing and engineering design problems, such as 3D printing, bridge construction, and aircraft engine production. A topology structure is used to describe how to allocate the material, *e.g.*, metal and concrete, in a designated spatial domain. Given the environmental input, *e.g.*, a pulling force or pressure, we want to identify the structure with the optimal property of interest, *e.g.*, maximum stiffness. Traditionally, we convert it to a constraint optimization problem, for which we minimize a compliance objective with a total volume constraint (Sigmund, 1997). Since the optimization often needs to repeatedly run numerical solvers to solve relevant PDEs, the computation is very expensive. We aim to learn a surrogate model with active learning, which can predict the optimal structure outright given the environmental input.

Specifically, our task is to design a linear elastic structure in an L-shape domain discretized in $[0, 1] \times [0, 1]$. The environmental input is a load at the bottom half right and described by two

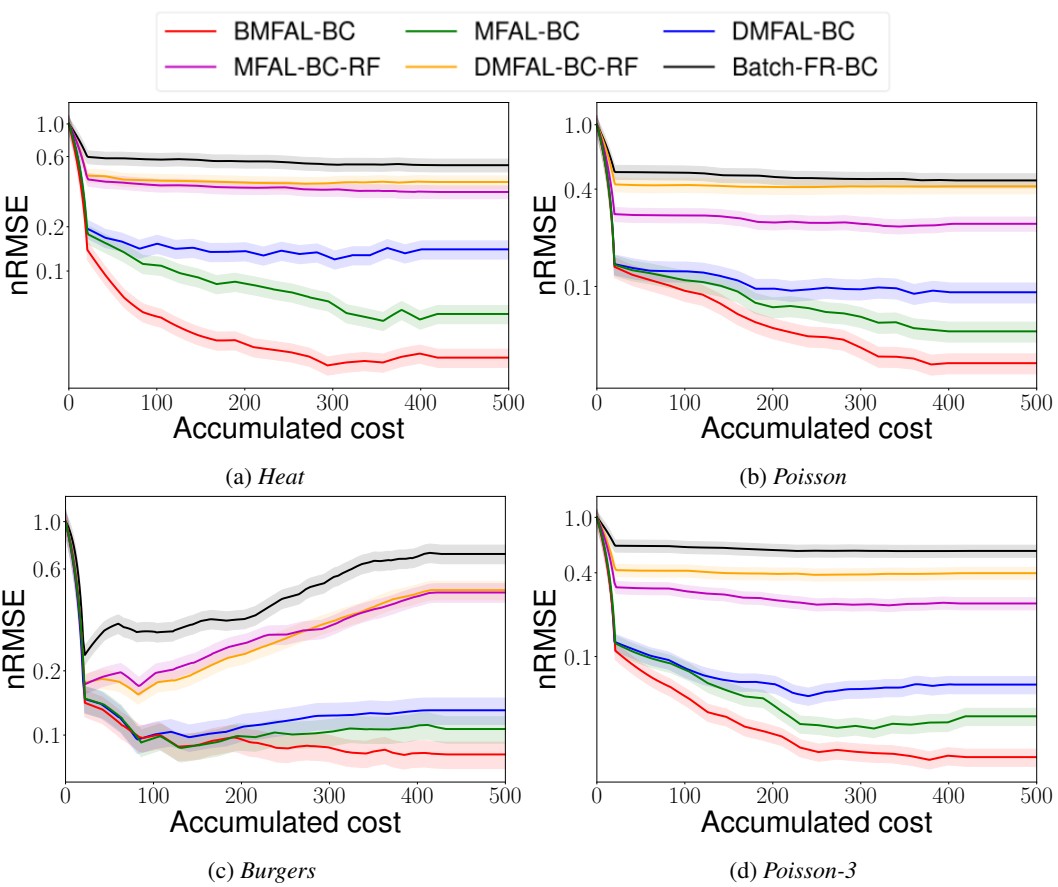

Figure 2: Normalized root-mean-square error (nRMSE) *vs.* accumulated data acquiring cost during batch multi-fidelity active learning, with budget 20 (normalized seconds) per batch. There are two fidelities in acquiring the examples in (a-c) and three fidelities in (d). The results were averaged over 5 runs. The shaded region shows the standard deviation.

parameters: angle (in $[0, \frac{\pi}{2}]$) and location (in $[0.5, 1]$). Given a particular load, we aim to find the structure that has the maximum stiffness. To calculate the compliance in the structure optimization, we need to repeatedly apply the Finite Element Method (FEM) (Keshavarzzadeh et al., 2018), where the fidelity is determined by the mesh. In the active learning, the training examples can be acquired at two fidelities. One corresponds to a $50 \times 50$ mesh used in the FEM, and the other $75 \times 75$. The output dimension of the two fidelities is therefore $2,500$ and $5,625$, respectively. The querying cost is the normalized average time to find the optimal structure (with the standard approach), $\lambda_1 = 1$ and $\lambda_2 = 3$. To evaluate the performance, 500 test structures were randomly generated with a $100 \times 100$ mesh. We interpolated the high-fidelity prediction of each method to the $100 \times 100$ mesh and then evaluated the prediction error.

At the beginning, we uniformly sampled the input (*i.e.*, load angle and location) to collect 10 structures at the first fidelity and 2 at the second fidelity, as the initial training set. We then ran all the active learning methods, with budget 20 per batch, to acquire 25 batches of examples. We examined the average nRMSE along with the accumulated cost of acquiring the examples. The results are reported in Fig. 3a. We can see that the prediction accuracy of BMFAL-BC is consistently better than all the competing methods during the active learning. The improvement becomes larger when more examples are acquired. The results confirm the advantage of BMFAL-BC.

### 5.3 Predicting Spatial-Temporal Fields of Flows

Third, we evaluated BMFAL-BC in computational fluid dynamics (CFD). We considered a classical application where the flow is inside a rectangular domain (represented by $[0, 1] \times [0, 1]$), and driven

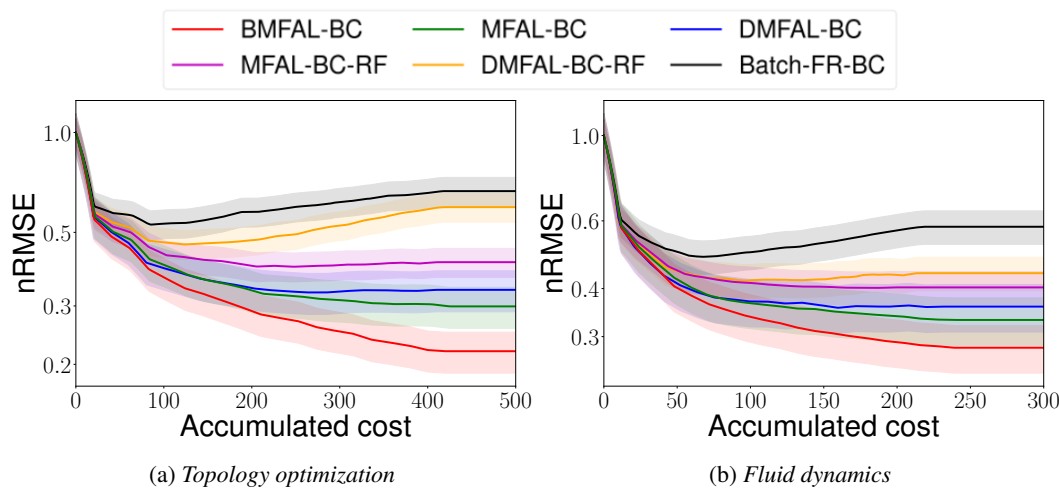

(a) *Topology optimization*  (b) *Fluid dynamics*

Figure 3: nRMSE *vs.* the accumulated cost in learning to predict topological structures and fulid dynamics. The budget was set to 20 and 10 for (a) and (b), respectively.

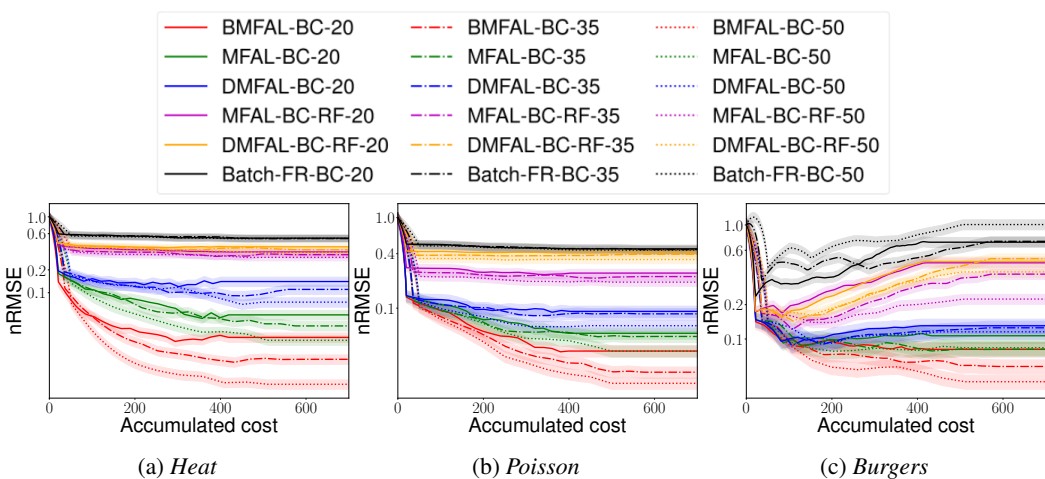

(a) *Heat*  (b) *Poisson*  (c) *Burgers*

Figure 4: nRMSE *vs.* the accumulated cost under different budgets per batch: $B \in \{20, 35, 50\}$.

by rotating boundaries with a constant velocity (Bozeman and Dalton, 1973). The velocities of different parts inside the flow will vary differently, and eventually lead to turbulent fluids. To compute the velocity field in the time spatial domain, we need to solve the incompressible Navier-Stokes equations (Chorin, 1968), which is known to be computationally challenging, especially under large Reynolds numbers. We considered the active learning task of predicting the first component of the velocity field at 20 evenly spaced points in the temporal domain $[0, 10]$. The training examples can be queried at two fidelities. The first fidelity uses a $50 \times 50$ mesh in the $[0, 1] \times [0, 1]$ spatial domain, and the second fidelity $75 \times 75$. Accordingly, the output dimensions are $50,000$ and $112,500$; the cost per query is $\lambda_1 = 1$ and $\lambda_2 = 3$. The input is five dimensional, including the velocities of the four boundaries and the Reynold number. The details are given in (Li et al., 2022). For testing, we computed the solution fields of 256 random inputs, using a $128 \times 128$ mesh. We used the cubic-spline interpolation to align the prediction made by each method to the $128 \times 128$ grid, and then calculated normalized RMSE. To conduct the active learning, we randomly generated 10 and 2 examples in each fidelity as the initial training set. We set the budget to 10, and ran each method to acquire 25 batches. We repeated the experiment for five times, and examined how the average nRMSE varied along with the accumulated cost. As shown in Fig. 3b, BMFAL-BC keeps exhibiting superior predictive performance during the course of active learning. Again, the more examples acquired, the more improvement of BMFAL-BC upon the competing methods. The results are consistent with the previous experiments. Note that throughout these comparisons, we focus on the accumulated

cost of querying (or generating) new examples, because it dominates the total cost, and in practice is the key bottleneck in learning the surrogate model. For example, in topology optimization (Sec 5.2) and the fluid dynamic experiment, running a high-fidelity solver once takes 300-500 seconds on our hardware, while the surrogate training takes less than 2 seconds, and our weighted greedy optimization of the batch acquisition function (Algorithm 1) takes a few seconds. One can imagine for practical larger-scale problems, the simulation cost, *e.g.*, taking hours or even days to generate one example, can be even more dominant and decisive.

### 5.4 Influence of Different Budgets

Finally, we examined how the budget choice will influence the performance of active learning. To this end, we varied the budget $B$ in $\{20, 35, 50\}$ and tested all the methods for Poisson's, Burger's and heat equations. We used the same two-fidelity settings as in Sec. 5.1. For each budget, we ran the experiment for five times. We show the average nRMSE *vs.* the accumulated cost in Fig. 4. As we can see, the larger the budget per batch, the better the running performance of BMFAL-BC. This is reasonable, because a larger budget allows our method to generate more queries in each batch and in the meantime to account for their correlations or information redundancy. Accordingly, the acquired training examples are overall more diverse and informative. Again, BMFAL-BC outperforms all the competing methods under every budget. The improvement of BMFAL-BC is bigger under larger budget choices. This together demonstrates the advantage of batch active learning that takes into account the correlations between queries.

## 6 Conclusion

We have presented BMFAL-BC, a budget-aware, batch multi-fidelity active learning algorithm for high-dimensional outputs. Our weighted greedy algorithm can efficiently generate a batch of input-fidelity pairs for querying under the budget constraint, without the need for combinatorially searching over the fidelities, while achieving good approximation guarantees. The results on several typical computational physical applications are encouraging.

## Acknowledgments

This work has been supported by MURI AFOSR grant FA9550-20-1-0358 and NSF CAREER Award IIS-2046295. JP thanks NSF CDS&E-1953350, IIS-1816149, CCF-2115677, and Visa Research.

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
