# Appendix

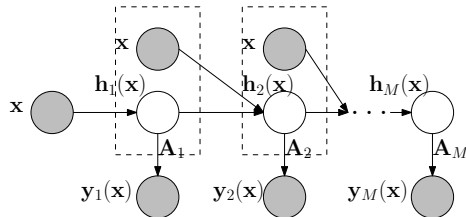

Figure 1: Graphical representation of DMFAL. The low dimensional latent output in each fidelity $\mathbf{h}_m(\mathbf{x})$ ($1 \leq m \leq M$) is generated by a (deep) neural network.

We now formally prove our main theoretical results on the approximate optimization properties of the Weighted-Greedy algorithm that we have proposed. In particular, these bounds are relative to the optimal algorithm with a budget $B$, we denote its mutual information as OPT($B$). We note that the optimal is with respect to the measurement of $\frac{1}{A}\sum_{l=1}^{A} \mathbb{I}(\mathcal{Y}_k, \mathbf{y}_M(\mathbf{x}_l')|\mathcal{D})$ on the $A$ Monte Carlo samples, and only over the space of inputs $\Omega$ and fidelities $\mathcal{M}$ we consider. If more Monte Carlo samples are considered, or somehow mutual information is computed precisely, or more fidelities are searched over, then the OPT($B$) considered will increase, and the near-optimality of the greedy algorithm will continue to be approximately proportional to that optimal potential value. Since DMFAL can actively choose an optimal $\mathbf{x} \in \Omega$ for a fixed fidelity $m$, which is already optimized over a continuous space, the optimal bound we consider OPT($B$) is relative to this method.

We now restate and prove the main results.

**Theorem 0.1** (Theorem **??**). *At any step of Weighted-Greedy (Algorithm **??**) before any choice of fidelity would exceed the budget, and the total budget used to that point is $B' < B$, then the mutual information of the current solution is within $(1 - 1/e)$ of OPT($B'$).*

*Proof.* Given a set of elements $\tilde{\Omega}$ and a submodular objective function $\phi$, it is well known that if one greedily selects items from $\tilde{\Omega}$ that most increase $\phi$ at each step, then after $t$ steps, the selected set achieves an objective value in $\phi$ within a $(1-1/e)$-factor of the optimal set of $t$ elements from $\tilde{\Omega}$ (Krause and Golovin, 2014). Our objective $\frac{1}{A}\sum_{l=1}^{A} \mathbb{I}(\mathcal{Y}_k, \mathbf{y}_M(\mathbf{x}_l')|\mathcal{D})$, where the mutual information is a classic submodular function (Krause and Guestrin, 2005). However, in our setting each item (an input-fidelity pair $(\mathbf{x}, m)$) has a cost $\lambda_m$ that counts against a total budget $B$. Our proof will convert this setting back to the classic unweighted setting so we can invoke the standard $(1-1/e)$-result.

Our Weighted-Greedy algorithm instead chooses an $(\mathbf{x}, m) \in \Omega \times \mathcal{M}$ to optimize $\hat{a}_{k+1} = \Delta_{\mathbf{x},m}/\lambda_m$ where $\Delta_{\mathbf{x},m} = \mathbb{I}(\mathcal{Y}_k \cup \{\mathbf{y}_m(\mathbf{x})\}, \mathbf{y}_M(\mathbf{x}_l')|\mathcal{D}) - \mathbb{I}(\mathcal{Y}_k, \mathbf{y}_M(\mathbf{x}_l')|\mathcal{D})$ is the increase in mutual information by adding $(\mathbf{x}, m)$. By scaling this $\Delta_{\mathbf{x},m}$ value by $1/\lambda_m$ we can imagine splitting the effect of $(\mathbf{x}, m)$ into $\lambda_m$ copies of itself, and considering each of these copies as unit-weight elements. We next argue that our

Weighted-Greedy algorithm will achieve the same result as if we split each item into $\lambda_m$ copies, and that the process on these copies aligns with the standard setting.

First lets observe Weighted-Greedy will achieve the same result as if each $(\mathbf{x}, m)$ was split into $\lambda_m$ copies. When we split each $(\mathbf{x}, m)$ into copies, each maintains the same scaled contribution of $\Delta_{\mathbf{x},m}/\lambda_m$ to our objective function. And we greedily add the item with largest contribution. So if some $(\mathbf{x}, m)$ has the largest contribution $\hat{a}_{k+1}$ in the weighted setting, then so will its unit weight copy in the unweighted setting. In the unit weight setting, after we add the first copy, this may effect the $\Delta_{\mathbf{x}',m'}/\lambda_{m'}$ contribution of some items $(\mathbf{x}', m') \in \mathcal{Q}_k$. By submodularity, all such items have diminishing returns and their contribution cannot increase. However, the unit weight copies of $(\mathbf{x}, m)$ are essentially independent, and so their $\Delta_{\mathbf{x},m}/\lambda_m$ score does not decrease (if we add all $\lambda_m$ we increase mutual information by a total of $\Delta_{\mathbf{x},m}$). Since no other item can increase its score, and the copies scores do not decrease, if they were selected for having the maximal score, they will continue to have the maximal score until they are exhausted. Hence, if we select one unit weight copy, we will add all of them consecutively, simulating the effect of adding the single weighted $(\mathbf{x}, m)$ at total cost $\lambda_m$. Note that by our assumption in the theorem statement, we can always add all of them.

Finally, we need to argue that this unit-weight setting can invoke the submodular optimization approximation result. For integer $\lambda_m$ and $B$ values, then this unit-weight version runs a submodular optimization with $B' < B$ steps. The acquisition function used to determine the greedy step is $\hat{a}_{k+1} = \Delta_{\mathbf{x},m}/\lambda_m$, but since we have divided each item $(\mathbf{x}, m)$ into unit weight components with independent contribution to the mutual information $\frac{1}{A}\sum_{l=1}^{A} \mathbb{I}(\mathcal{Y}_k, \mathbf{y}_M(\mathbf{x}'_l)|\mathcal{D})$ they satisfy submodularity. Then the weight is the same among all items so it can be ignored, and it maps to the standard submodular optimization with $B'$ steps, and achieves within $(1-1/e)$ of OPT($B'$) as desired. $\square$

**Corollary 0.1** (Corollary **??**). *If Weighted-Greedy (Algorithm **??**) is run until input-fidelity pair $(\mathbf{x}, m)$ that corresponds with the maximal acquisition function $\hat{a}_{k+1}(\mathbf{x}, m)$ would exceed the budget, it selects that input-fidelity pair anyways (the solution exceeds the budget $B$) and then terminates, the solution obtained is within $(1-1/e)$ of OPT(B).*

*Proof.* Consider that the extended Weighted-Greedy algorithm terminates using total $B^+ \geq B$ total budget. By Theorem **??**, if we had $B^+$ budget, then this would achieve within $(1-1/e)$ of OPT($B^+$). And since OPT($B^+$) $\geq$ OPT($B$), then this is within $(1-1/e)$ of OPT($B$) as well. $\square$

These results imply that the Weight-Greedy algorithm achieves the desired $(1-1/e)$-approximation until we are near the budget, or we slightly exceed it. If the maximal weight item $\lambda_M$ is close to the full budget, then we are always in this unbounded case – or may need to greatly exceed the budget to obtain a guarantee. However, on the other hand, if $\lambda_M$ is fixed and the budget $B$ increases, then our bounds become more accurate. In either case we can obtain a score within $(1 - \frac{\lambda_M}{B})(1 - 1/e)$ of the OPT at a budget $B$ – by excluding the part where the greedy choice may exceed the budget. So as $\lambda_M/B$ goes to 0, then the approximation goes to $(1 - 1/e)$.

While we have proven these results in the context of the specific approximated mutual information and parameter space $\Omega \times \mathcal{M}$ these nearly $(1 - 1/e)$-optimal results

will apply to any submodular optimization function, scaled by its optomal cost with a budget $B$.

Note that Leskovec et al. (2007) proposed another approach to dealing with this budgeted submodular optimization. They proposed to run two optimization schemes, one the method we analyze, and one that simply chooses the items that maximize $\Delta_{\mathbf{x},m}$ at each step while ignoring the difference in their cost $\lambda_m$. They show that while the first one may not achieve $(1 - 1/e)$-approximation, one of these schemes must achieve that optimality. The cost of running both of them, however, is twice the budget, so in the worst case this combined scheme only achieves within $(1/2)(1 - 1/e)$ of the optimal. This run-twice approach is also wasteful in practice, so we focused on showing what could be proven (near $(1 - 1/e)$-approximation) of just Weighted-Greedy. In fact, as long as $\lambda_M/B \leq 1/2$, we already match their worst case bound.