# OpenReview forum: "Batch Multi-Fidelity Active Learning with Budget Constraints"
_NeurIPS.cc/2022/Conference — NeurIPS 2022 Accept_

### Official Review · Reviewer_v34g · 2022-07-07

**Rating:** 6
**Confidence:** 2
**Soundness:** 2 fair
**Presentation:** 2 fair
**Contribution:** 2 fair

**Summary:**

The paper presentes BMFAL-BC, a new batch multi-fidelity active learning approach that takes into account budget constraints. Differently from prior work, BMFAL-BC selects query points at different fidelities ensuring both that the queried points have low correlation among themselves and that the budget is constraint is complied with. The intuition behind ensuring low correlation among the queried points is that higher diversity will contribute to a better learning process, i.e., higher learning efficiency, hence decreasing the required number of queries and the cost of the learning process. The authors prove bounds for their algorithm and compare their approach against state-of the art multi fidelity active learning approaches, showing that BMFAL-BC finds better solutions at cheaper learning costs.

**Questions:**

**a)** I don't understand the meaning of "the cost of computing examples at each fidelity, was set to the normalized average running time". Does this mean you gathered queries from each fidelity multiple times, measured how long each query took, and set that as the cost? Also, "the normalized average running time" of what?

**b)** In the sentence "For active learning, we randomly generated {10, 2} and {10, 5, 2} training examples", what does the 10 and 2 stand for in {10, 2}? Same question for {10, 5, 2}.

**c)** When BMFAL-BC selects a batch, does it test all points in that batch? Also, do all points in a batch belong to the same fidelity?

**d)** If my understanding is correct, BMFAL-BC has a budget B per batch, which it can spend to evaluate several queries. In your implementation of DMFAL-BC do you allow it to select as many queries/points as it needs to reach budget B before you consider the batch as complete? Or do you terminate	 the run the first time DMFAL-BC is complete?

**e)** Is the accumulated cost equal to number of evaluated batches x B?

**f)** How many more queries does BMFAL-BC execute compared to DMFAL-BC?

**g)** What is the time complexity/overhead of the proposed method compared to existing methods?

**Comments:**
- PDE is introduced in line 224 however it is used already in line 25. Maybe introduce it the first time you use it for clarity.
- Lines 38-42 lack supporting references
- The sentence in lines 226-227 is not complete.
- The legend of Figures 2 and 3 could be made easier to read: all methods have BC in their names, so BC is redundant. Also, maybe you could align the methods with the same name, i.e., MFAL-BC would be above MFAL-BC-RF and DMFAL-BC above DMFAL-BC-RF
- typo in line 485? when --> well


**Limitations:**

No limitations were discussed. Under which conditions can the method be applicable? Do the underlying problems need to adhere to any specific format? Are there any underlying assumptions? For instance with respect to the shape of the function under optimisation?

**Strengths And Weaknesses:**

**Originality:** The authors propose a new method that extends existing work in order to account for budget constraints and thus decrease the cost of the learning process. This is a novel combination and application of existing techniques to the context of batch multi-fidelity active learning. As far as I know, active learning related work is adequately cited however there is significant (and recent) work in the context of multi-fidelity hyper-parameter tuning optimisation which is not mentioned. Specifically, the authors may find works such as [1] and [2] relevant for their research. [1] seems a particularly good citation for line 123.

**Quality:** The claims seem to be sound, however I did not carefully check the math and proofs. The evaluation methods are appropriate and consistent with those used by related work. The weaknesses and limitations could be discussed more explicitly. One possible limitation that is not discussed is the use of one NN for each fidelity. It would be interesting to explore research directions that address this overhead and use only one NN, for example.

**Clarity:** The active learning setting could be better described. It is not clearly explained how each batch of samples actively contributes to the learning. There could be more information on the experiments and parameters tested (in table format) in the appendix. This would make reproducibility easier. Also, some guidance on possible ways to set the budget could aid potential users of this approach.

**Significance:** The problem tackled in the paper is a relevant one and having approaches to optimize the search and learning process at lower costs is valuable for the community. The proposed method is better than existing ones on the evaluated scenarios.


[1] Mendes, Pedro, et al. "Trimtuner: Efficient optimization of machine learning jobs in the cloud via sub-sampling." 2020 28th International Symposium on Modeling, Analysis, and Simulation of Computer and Telecommunication Systems (MASCOTS). IEEE, 2020.

[2] Klein, Aaron, et al. "Fast bayesian optimization of machine learning hyperparameters on large datasets." Artificial intelligence and statistics. PMLR, 2017.

---

> ### Author Response · Authors · 2022-07-29
> **Thanks for your valuable and detailed comments.**
>
> Thanks for your valuable and detailed comments. Here are our responses. C: comments; R: response.
>
> C1: active learning related work is adequately cited but there is significant (and recent) work in the context of multi-fidelity hyper-parameter tuning optimisation not mentioned…
>
> R1: Thanks for providing these great references! We do agree “[1] is particularly good citation for line 123”. We will cite and discuss  these works [1][2] in our paper.
>
>
> C2:  Does “the cost of computing examples at each fidelity, was set to the normalized average running time"” mean …?  "the normalized average running time" of what?
>
> R2: Yes. We followed (Li et al., 2022) to get the cost for each fidelity, which comes from the average running time of multiple queries on the same hardware. “normalized” means that we normalize the average running time by that of the lowest fidelity. For example, if the average running time for 3 fidelities is (60s, 120s, 600s), the normalized average running time (i.e., cost) is (1, 2, 10). Thanks for the question. We will clarify this point in the paper.
>
>
> C3: In the sentence "For active learning, we randomly generated {10, 2} and {10, 5, 2} training examples", what does the 10 and 2 stand for in {10, 2}? Same question for {10, 5, 2}.
>
> R3: {10, 2} means we generated 10 examples of fidelity 1, and 2 examples of fidelity 2; similarly, {10, 5, 2} means we generated 10, 5 ,2 examples of fidelity 1, 2, and 3, respectively. Thanks for the question. We will rephrase the sentence to be more clear.
>
> C4: When BMFAL-BC selects a batch, does it test all points in that batch? Also, do all points in a batch belong to the same fidelity?
>
> R4: Yes, once the batch is identified, we will query (test) all the points to obtain a batch of examples. The query points do not necessarily belong to the same fidelity. Our method identifies both the input and the fidelity for each query point in the batch. The goal is to find a smart combination of multi-fidelity queries in each batch to best improve learning while respecting the budget. See Algorithm 1 (line 3), Eq6 and Sec 3.2 for details.
>
> C5: In your implementation of DMFAL-BC do you allow it to select as many queries/points as it needs to reach budget B before you consider the batch as complete? Or do you terminate the run the first time DMFAL-BC is complete?
>
> R5: Great question! DMFAL-BC follows the same budget constraint as in our method. That is, it "selects as many queries/points as it needs to reach budget B". The difference is that each step DMFAL-BC optimizes a single acquisition function (Eq.2), then acquires the example, adds it to the dataset, and retrains the model. It keeps doing this until the budget is used up. if DMFAL-BC queries 10 examples in a batch, it will retrain the model for 10 times. This is in contrast to our method that considers the joint effect of a batch of queries and optimizing a batch acquisition function (Eq4) that penalizes strong correlations within the batch queries (see line 136-139). Only when all the queries/points in the batch are identified, our method acquires the examples, adds them together into the dataset, and retrains the model. Hence our method only retrains the model once for each batch. Thanks for the question. We will highlight all these in our paper.
>
> C6: Is the accumulated cost equal to number of evaluated batches x B?
>
> R6: Great question! This is in general not true since we might have cases that the budget is not completely used up, but the remaining budget cannot afford any query. Then, the accumulated cost of this batch is less than B. Accordingly, the total accumulated cost is not #batches x B. For example, suppose we have two fidelities with costs 3 and 4. If we have B = 5, then the budget can never be used-up. In our experiments, since the cost of each fidelity is an integer and the smallest is 1, when B is an integer, the budget can always be filled up with queries of the smallest fidelity. So the accumulated cost is indeed #batches x B. However, when B is not an integer, the smallest cost is not 1, and/or any cost is not an integer, the conclusion will not hold.
>
>
> C7: How many more queries does BMFAL-BC execute compared to DMFAL-BC?
>
> R7: Great questions. Since both methods consider multi-fidelity queries (according to the running status of their own models), more queries do not necessarily give better learning performance. In fact, BMFAL-BC tends to query more higher-fidelity examples, and thereby #queries is smaller (under the same budget). For example, in Poission-3, BMFAL-BC and DMFAL-BC executed 12 and 15 queries on average while in Heat, 28 and 35. We will supplement the number of queries of each method in our paper.

---

> > ### Author Response · Authors · 2022-07-29
> > **Response continuing**
> >
> > C8: What is the time complexity/overhead of the proposed method compared to existing methods?
> >
> > R8: Great question. For simplicity, we omit the common factor that accounts for the internal optimization (e.g., LBFGS) and mutual information computation. The time complexity of DMFAL-BC, MFAL-BC, DMFAL-BC-RF, and MFAL-BC-RF in optimizing the acquisition function is $O(\frac{B}{\lambda_1} M)$, $O(A\frac{B}{\lambda_1} M)$,  $ O(\frac{B}{\lambda_1})$, and $O(A\frac{B}{\lambda_1})$, respectively. Here $B/\lambda_1$ is the maximum number of queries in a batch. As a comparison, the complexity of our method is $O(A\frac{B}{\lambda_1}M)$. Note that we explicitly write out the constant A (#samples in computing our new acquisition function in Eq5) to highlight the difference with the other methods. Batch-FR-BC has no cost in optimizing the acquisition function, because it randomly samples both the input location and fidelity.  Suppose the data size is N. Since the model training is conduced by running SGD to pass the data for a maximum #epochs (see (Li et al., 2022)), the training  complexity is O(N). Then for a batch, the training time complexity of DMFAL-BC, MFAL-BC, DMFAL-BC-RF, MFAL-BC-RF is $O(\sum_{j=1}^{B/\lambda_1} (N+j)) = O\left(N \frac{B}{\lambda_1} + \left(\frac{B}{\lambda_1}\right)^2\right)$, because they sequentially add each acquired example and then retrain the model. Our method and Batch-FR-BC only retrain once after adding the acquired examples altogether to the dataset, so the complexity is $O(N + \frac{B}{\lambda_1})$. Despite the difference, the training and acquisition function optimization of each method only takes a few seconds in our experiments, and the total cost is dominated by the labeling cost. See R1 to Reviewer SBrH. We will supplement the analysis and comparison of the overhead in our paper.
> >
> >
> >
> > C9: Other comments on paper writing.
> >
> > R9: We appreciate these detailed suggestions. We will polish our paper and add the references accordingly to further improve our paper.
> >
> > C10: No limitations were discussed.
> >
> > R10: Actually, we have mentioned the limitations in the paper. Our greedy algorithm can potentially degrade when the budget is about to fill up. So our method achieves a near (1-1/e) approximation; see line 165-169, Appendix and R3 to reviewer Uvz4. In the future work, we will seek to adjust the strategy when the budget is about to fill up so as to avoid the degradation. Note that our method does not limit the function types/shapes in active learning and can be used in any scientific and engineering area that needs physical simulations yet have computational resource limits.  We will add more detailed discussion about the limitation of our work, possible solutions and plan for the future work.

---

> > > ### Comment · Reviewer_v34g · 2022-08-08
> > > **Thank you for the clarifications**
> > >
> > > I would like to thank the authors for the clarifications. I have no further questions.

---

### Official Review · Reviewer_Uvz4 · 2022-07-09

**Rating:** 6
**Confidence:** 3
**Soundness:** 3 good
**Presentation:** 3 good
**Contribution:** 3 good

**Summary:**

Multi-fidelity learning is applied whenever observations generated from a cheap but less precise surrogate model are combined with limited observations from the actual process, such that a mapping can be learned between low and higher fidelity outputs. This is particularly challenging when the output dimensionality becomes greater at higher fidelities. In this work, the authors focus on the problem of deciding at which fidelity and location of the input domain the next sample points should be acquired. In order to mitigate the risk of successive sequential acquisitions focusing on the same limited region, the authors propose a batch active learning scheme which jointly selects different input points to sample across different fidelities, subject to a computational budget. Via a range of experiments, the authors show how the selection procedure results in more efficient learning than other competing methods when given with the same sampling budget.

**Questions:**

My largest reservations around this work lie around the lack of citation and connection to _“Batch Multi-Fidelity Bayesian Optimization with Deep Auto-Regressive Networks”_, which shares a lot of the motivation and approach of the work presented here. Again, while I appreciate that the two use-cases have different requirements, the overall significance and novelty of this paper come across as slighter in light of this other work.


**Limitations:**

I appreciated the discussion on the quality of the weighted greedy optimisation procedure, which was also appropriately hinted at in the introduction of this work. As mentioned earlier, providing an anecdotal example of where this could occur (even if synthetic) would be helpful.

There should not be any ethical or societal issues tied to the publication of this work.


**Strengths And Weaknesses:**

**Strengths**

_[Significance]_ - As evidenced by the variety of application domains featured in the experiments section, multi-fidelity modelling is an emerging topic in many practical areas. This paper tackles an interesting problem within this space that does not appear to have been directly investigated for multi-fidelity regression prior to this work.

*[Clarity]* - The paper is well-written overall and easy to follow. There are a few typos here and there, but these aren’t distracting and should be easily fixed in a second revision.

*[Evaluation]* - I appreciated how the experiments were varied by leveraging datasets sourced from different domains. The choice of methods used for comparison is also appropriate, and the plots effectively illustrate both the overall performance and variance of the methods. The results indicate that the proposed approach consistently performs better than competing methods, which adds credence to the usefulness of the contributions presented in this work.

**Weaknesses**

*[Novelty]* - The paper bears similarities to another recent paper, _“Batch Multi-Fidelity Bayesian Optimization with Deep Auto-Regressive Networks”_, which shares a lot of the motivation and set-up for this work, with the exception that it is centred on the single-output Bayesian optimisation case (rather than the high-dimensional outputs considered here). While I appreciate that the two set-ups are different, there is considerable overlap which makes the lack of any reference to that paper particularly puzzling. I would like to see this featured more prominently in a future revision, as it is currently conspicuous by its absence.

*[Evaluation]* - One disappointing aspect about the evaluation section is that most of the text is devoted to explaining the datasets and the parameters used for the different methods, whereas the insights and findings themselves are limited to one or two brief sentences at the end of each experiment. While the results of the graphs are fairly self-evident, further insights or anecdotal examples of where the batch selection is particularly effective would be welcome.

*[Completeness]* - One interesting limitation highlighted in the presentation of the method is the potential degradation of the greedy optimisation mechanism - however, this isn’t revisited in the experimental evaluation, whereas it would add to the paper’s completeness if the authors could include an example of how/where such behaviour manifests itself.

---

> ### Author Response · Authors · 2022-07-29
> **Thanks for your valuable comments.**
>
> Thanks for your valuable comments. Here are our responses. C: comments; R: response.
>
> C1: lack of citation and connection to another recent paper, “Batch Multi-Fidelity Bayesian Optimization with Deep Auto-Regressive Networks”
>
>
> R1: Thanks much for bringing up the reference. We will definitely cite and discuss this work in our paper. Note that the prior paper (Li et al., 2022), which we extend to the proposed batch multi-fidelity (MF) active learning (AL), has provided a detailed discussion about the works in MF Bayesian optimization (BO), and their connection/difference with MF AL (see Sec 5 “Related Work” in their paper). From the high-level view, both the mentioned paper and our work employ a similar interleaving procedure, i.e., determining a new batch of queries by optimizing an acquisition function, issuing the queries to collect new examples, and updating the surrogate model. However, both the learning settings and acquisition functions are different, e.g., we consider the budget constraint while the mentioned MF-BO work does not. Thereby, the computation and optimization techniques of the two works are totally different. The MF-BO paper just uses HMC samples (of the single outputs) and sample covariance matrices to compute the acquisition functions, while our method and (Li et al., 2022) use the multi-variate delta method and matrix identities to overcome the challenge of the high-dimensional outputs; the MF-BO paper uses alternating optimization to search over multiple fidelities, while our work develops a weighted greedy algorithm with additional theoretical guarantees. We will highlight these differences and connections in our paper.
>
> C2: While the results of the graphs are fairly self-evident, further insights or anecdotal examples of where the batch selection is particularly effective would be welcome.
>
>
> R2: Thanks much for the suggestion. That’s really a great idea. We will supplement more detailed discussions about the insights of the results and provide the additional anecdotal examples as you suggested.
>
>
> C3: One interesting limitation highlighted in the presentation of the method is the potential degradation of the greedy optimization mechanism … providing an anecdotal example of where this could occur (even if synthetic) would be helpful.
>
> R3: Great idea! Such degradation can happen when the budget is about to fill up. There is a possibility that some query with a small benefit yet higher benefit/cost ratio is first selected and occupies the budget, causing no budget left for a subsequent query with higher benefit. We will add such examples to showcase the potential degradation.

---

> > ### Comment · Reviewer_Uvz4 · 2022-08-07
> > **Acknowledgement of Rebuttal**
> >
> > Thank you for your responses to the feedback provided across all reviews, especially for the clarifications on the connections to the highlighted paper.
> >
> > In view of the feedback shared by other reviewers, I am currently inclined to leave my score unchanged due to the scope for several incremental improvements to the paper that would increase its overall completeness. However, I will participate in any further internal discussions on the paper and update my score accordingly.

---

### Official Review · Reviewer_UwmX · 2022-07-11

**Rating:** 7
**Confidence:** 2
**Soundness:** 4 excellent
**Presentation:** 3 good
**Contribution:** 3 good

**Summary:**


The paper proposes BMFAL an extension of DMFAL for batch multi-fidelity active learning with budget constraints. The proposed method is a greedy sequential algorithm that selects datapoints which maximise the information gain. The experiments show that BMFAL outperform a naive extension of DMFAL in the batch multi-fidelity active learning with budget constraints.

**Questions:**


 - I think Section 3.3 could be improved. The algorithm complexity in 3.3 ignores the cost of searching (i.e., performing the argmax) $x \in \Omega$. In addition, it ignores the cost of computing the acquisition function which is not negligible. The space complexity can also be simplified to $O(\frac{B}{\lambda_1}(r+d))$ since the constant is negligible.
 - Figure 2 is unnecessarily large. The figure can be made smaller and further discussion regarding the limitations of the method should be included.

**Limitations:**

The authors could include further discussion regarding in what specific areas batch multi-fidelity active learning with budget constraints is used. The discussion should include the potential negative societal impact of those areas.

**Strengths And Weaknesses:**


Strengths:
 - To my knowledge, this is the first method proposed for Batch Multi-Fidelity Active Learning with Budget Constraints. The setting is very specific but it's an important setting. Although not mentioned in the paper, multi-fidelity methods are also very useful for biological sequence design or drug design. In these settings, wet-lab experiments are very expensive, so there are methods to approximate them at various fidelities.
 - The experiments are conclusive and show a clear benefit of BMFAL over DMFAL in the batch setting.

Weaknesses:
 - The experiments are limited to up to 3 fidelities. However, I would imagine that in certain settings we would want a larger number of fidelities. It would be interesting to see how this scales to a larger number of fidelities.
 - No code is included but the details appear sufficient to reproduce the experiments.


Overall, I would rate the paper:

Novelty: Medium

Clarity: High

Significance: Medium-High

---

> ### Author Response · Authors · 2022-07-29
> **Thanks for your valuable comments**
>
> Thanks for your valuable comments. Here are our responses. C: comments; R: response.
>
> C1: The experiments are limited to up to 3 fidelities.
>
> R1: Great suggestions. We just followed the prior work (Li et al., 2022) to use 2 or 3 fidelities for experimental validation.  We will supplement the results with more fidelities.
>
> C2: No code is included but the details appear sufficient to reproduce the experiments.
>
> R2: Thanks for your comments. We will release our implementation, experimental data and simulation code in Github.
>
>
> C3: Section 3.3 could be improved… The algorithm complexity in 3.3 ignores the cost of searching and the cost of computing the acquisition function …. The space complexity can also be simplified since the constant is negligible.
>
> R3: Great suggestion. We do agree. For simplicity, we omit the cost terms for searching and acquisition function computation, which are detailed in L-BFGS paper and prior work (Li et al., 2022). These should be multiplied with the complexity term in our paper. We will supplement them and also remove the constant in space complexity.
>
> C4: Figure 2 is unnecessarily large. The figure can be made smaller and further discussion regarding the limitations of the method should be included.
>
> R4: Great suggestion, we will reduce the size of Fig. 2 and supplement the discussion accordingly.
>
> C5: The authors could include further discussion regarding in what specific areas batch multi-fidelity active learning with budget constraints is used. The discussion should include the potential negative societal impact of those areas.
>
> R5: Great suggestion! We will supplement such discussions. Our method can be used in any scientific and engineering area that requires physical simulations and have computational resource limits, such as the design and manufacturing of aircrafts and cars, weather forecast, and bridge design, building energy management, etc. It can also be used in biological sequence design or drug design (as you mentioned). However, if our method is used in the development of fatal weapons, it might bring negative societal impacts.

---

> > ### Comment · Reviewer_UwmX · 2022-08-06
> > **Thanks for the clarifications.**
> >
> > I would like to thank the authors for their clarifications.

---

### Official Review · Reviewer_SBrH · 2022-07-12

**Rating:** 6
**Confidence:** 3
**Soundness:** 3 good
**Presentation:** 4 excellent
**Contribution:** 4 excellent

**Summary:**

This paper proposes a novel batch active learning algorithm for multi-fidelity tasks. While prior work only queries one example at a time, the authors propose novel mutual information objectives that also take into account of the correlation within a batch. Moreover, the authors provide a greedy relaxation algorithm in solving the optimization, which is further shown theoretically to be an (1 - 1/e)-approximation of the optimal objective. Extensive experiments have been conducted on four physics applications while outperforming the existing baseline significantly.

**Questions:**

1. Is knowing a-priori the costs lambda_m for each fidelity m a reasonable assumption? Does it make sense to assume they are fixed costs?
2. Other issues mentioned in the weakness section.

**Limitations:**

1. An advantage of batch active learning is the fact we could parallelize annotation. Now, consider if the cost of annotation comes only from time constraints, then the annotation cost of a batch actually comes from the example that takes the longest to annotate (highest fidelity) rather than the sum of the costs from each fidelity. It is not immediate to me that the method proposed in this paper is sufficient to solve such settings. Therefore, I recommend the authors to properly address as a limitation in their paper.

**Strengths And Weaknesses:**

Strengths:
1. The paper targets an important issue not addressed by prior work, which is the diversity among collected examples. The objectives are straight-forward and very intuitive.
2. The algorithm appears to be rather efficient and also enjoys theoretical worst case guarantees.
3. The experiments clearly shows the proposed algorithm's advantage in the batch setting.

Weaknesses:
1. The high labeling cost incurred by physical simulations motivated this work on applying active learning. However, when addressing accumulated cost, it does not seem to take into account the extra computation cost of the active learning algorithm. In the time complexity section, it is also lacking a comparison with existing work.
2. Moreover, a comparison with uniform random sampling strategies may be informative, especially when taking into account the extra computation cost incurred by the active learning algorithm. If they significantly underperforms, I think it's worth addressing in the text at least.


Minor Suggestion:
1. On the last line of equation (6), it's better to put sum_{hat{m}} lambda_{hat{m}} into parentheses.

---

> ### Author Response · Authors · 2022-07-29
> **Thanks for your valuable and insightful comments**
>
> Thanks for your valuable and insightful comments. Here are our responses. C: comments; R: response.
>
> C1: When addressing accumulated cost, it does not seem to take into account the extra computation cost of the active learning algorithm… lack a comparison in time complexity analysis…a comparison with uniform random sampling strategies may be informative, especially when taking into account the extra computation cost incurred by the active learning algorithm…
>
> R1: Great comments and suggestions! We focus on the accumulated labeling cost, because it dominates the total cost, and in practice is the key bottleneck in learning the surrogate model. For example, in topology optimization and fluid dynamics experiment (Sec 5.2&5.3), running a high-fidelity solver once takes 300-500 seconds on our hardware, while the surrogate training takes less than 2 seconds, and our weighted greedy optimization of the batch acquisition function takes a few seconds. One can imagine for practical larger-scale problems, the simulation (labeling) cost, e.g., taking hours or days to generate one example, can be even more dominant and decisive. However, we do agree that considering the other cost can give more thorough and convincing evaluation of our method. Thanks for the great suggestion. We will supplement a detailed comparison that accounts for the other costs, i.e., the surrogate model training and optimization of the acquisition function, to enhance our paper and enrich the results. We will also list the time complexity of the competing methods in the paper (see R8 for Reviewer v34g).
>
>
> C2: On the last line of equation (6), it's better to put sum_{hat{m}} lambda_{hat{m}} into parentheses.
>
> R2: Great suggestion. We will modify the form of eq6 accordingly.
>
>
> C3: Is knowing a-priori the costs lambda_m for each fidelity m a reasonable assumption? Does it make sense to assume they are fixed costs?
>
> R3: Great question. We follow the prior work (Li et al., 2022) to use a fixed cost in each fidelity. We believe this setting is reasonable, because the fidelity is determined by the mesh size or finite element length, which determines the scale of the linear system (e.g., the number of variables and time steps) after discretization and the complexity of solving the system. From both the analysis and test runs, it is easy to know the cost lambda_m a priori.
>
>
>
> C4: An advantage of batch active learning is the fact we could parallelize annotation. Now, consider if the cost of annotation comes only from time constraints, then the annotation cost of a batch actually comes from the example that takes the longest to annotate (highest fidelity) rather than the sum of the costs from each fidelity. It is not immediate to me that the method proposed in this paper is sufficient to solve such settings. Therefore, I recommend the authors to properly address as a limitation in their paper.
>
> R4: Thanks for the insightful comments. In this work, we consider the total cost of all the queries, because in many applications, e.g., cloud computing or high-performance computing, even we can submit jobs in parallel, the total expense is often accounted, because each job still consumes resources. But we do agree, if we only view the longest run as the cost, we will need to revise our algorithm, maybe substantially. For example, the budget constraint in Eq6 needs to be changed and the greedy algorithm might need to be redeveloped. We will definitely acknowledge and discuss this issue in our paper.

---

> > ### Comment · Reviewer_SBrH · 2022-08-03
> > **Thanks for the clarifications**
> >
> > I would like to thank the authors for their clarifications. As the authors acknowledge, I hope some of these discussions could be included in the camera-ready version of your paper.

---

### Meta-Review · Area_Chair_ni1a · 2022-08-26

**Recommendation:** Accept
**Confidence:** Certain

**Metareview:**

The paper proposes a new batch active learning algorithm for multi-fidelity tasks. The proposal is a sequential greedy algorithmwith approximation guarantees and authors provide extensive experimental comparison.

The reviewers generally agree that the paper should be accepted.

In the camera-ready version, the authors are strongly encouraged to include:
- additional discussions that they promised in their rebuttal
- additional experimental results with more than 3 fidelities
- a pointer to their open-sourced code and datasets

**Award:**

No

---

### Decision · Program_Chairs · 2022-09-14

Accept